# Exploring factors contributing to patient decision-making in the care journey to elective hernia care in Kenya

Helen W. Li[1,2,3,4]*◉, Jesse Kiprono Too[4]◉, Sarah Nyanchama Nyariki[4,5]◉,
Charles Nathan Nessle[3,6‡], Sara Malone[7,‡], Rachel Matsumoto[7‡],
Teddy Ashibende Aurah[4‡], Jeffrey A. Blatnik[1‡], JoAnna Hunter-Squires[2,5]◉,
Ivan Seno Saruni[5,8]◉

1 Department of Surgery, Washington University in St Louis, St Louis, Missouri, United States of America, 2 Department of Surgery, Indiana University School of Medicine, Indianapolis, Indiana, United States of America, 3 Fogarty LAUNCH fellow, Northern Pacific Global Health Fellows Program, Fogarty International Center of the National Institutes of Health, Bethesda, Maryland, United States of America, 4 Department of Surgery, Moi University School of Medicine, Eldoret, Kenya, 5 Academic Model Providing Access to Healthcare (AMPATH), Eldoret, Kenya, 6 Department of Pediatrics, Division of Pediatric Hematology/Oncology, University of Michigan Medical School, Ann Arbor, Michigan, United States of America, 7 Institute for Public Health, Washington University in St Louis. St Louis, Missouri, United States of America, 8 Department of Surgery, Moi Teaching and Referral Hospital, Eldoret, Kenya

◉ These authors contributed equally to this work.
‡ These authors also contributed equally to this work.
* helenli@wustl.edu

## Abstract

### Background

Capacity for elective general surgical care is an important reflection of a health system's ability to meet a population's surgical needs and is currently known to be inadequate in many low- and middle-income countries. Patient agency is a key, understudied factor which shapes how and when patients ultimately decide to engage with formal care. Understanding factors which influence patient care seeking activity can have important implications for how current and future health systems may be utilized. This study aims to explore how patients approach the navigation and triage of their elective hernia condition within the Kenyan surgical care system.

### Methods

We conducted a qualitative study of 38 convenience-sampled patients diagnosed with an elective hernia condition at a tertiary referral hospital in Kenya between November 2023 and March 2024. We utilized Braun and Clarke's six-step model of thematic analysis to generate key themes across the phases of care seeking, reaching and receiving as modeled in the Three Delays Framework.

**Data availability statement:** All relevant data are within the manuscript and its Supporting Information files.

**Funding:** Research reported in this publication was supported by the Fogarty International Center of the National Institutes of Health under grant #D43TW009345 awarded to the Northern Pacific Global Health Fellows Program. The content is solely the responsibility of the authors and does not necessarily represent the official views of the National Institutes of Health. The funders had no role in study design, data collection and analysis, decision to publish, or preparation of the manuscript.

**Competing interests:** The authors have declared that no competing interests exist.

## Results

We identified three main cross-cutting themes including (1) the flow of power from patients to providers, and vice versa, take the form of consent or knowledge, respectively; (2) trust is a limited currency required for patients to engage with formal care; and (3) internal and external contextual factors remain the foundation for patient-provider care activities. We incorporated these themes together in a framework which illustrates the cyclical nature by which each factor feeds back on the others, ultimately affecting patient care.

## Conclusions

Fluctuating flows of patient power and trust interacts with existing infrastructural context to influence the ability of a health system to generate care. Recognizing the interaction of these key factors may have important bearing on the successful implementation of any larger systemic efforts or policies to improve access to elective surgical care.

## Introduction

Elective general surgical care has a fragile and often leaky pipeline, a phenomenon highlighted following the COVID-19 pandemic when an estimated 82% of elective surgeries for benign conditions were canceled or postponed globally [1,2]. Elective surgical capacity can reflect a health system's ability to meet the surgical needs of its population in a timely fashion, particularly given that managing surgical needs in an elective manner avoids emergent surgeries which are known to be associated with worsened patient outcomes. Initiatives like National Surgical, Obstetric and Anesthesia Plans (NSOAPs) or the Global Alliance for Surgical, Obstetric, Trauma, and Anesthesia Care (G4 Alliance) 11 Best Practice Recommendations on the optimal organization of surgical services seek to integrate operative care into broader country-level health systems strengthening plans [3,4]. While these system-level approaches are critical for coordinating the expansion of access to elective surgical care, a gap remains in considering how patients may interact with these well-laid plans.

The intrinsic factor of choice present in elective surgery introduces complex care triaging decisions for providers and patients which can affect the timing of surgical care. Hernia, an abnormal protrusion of intra-abdominal contents through a weakness in the abdominal wall, is a unique pathology which must balance the timing of a simple, curative, elective surgery with the unpredictable risk of developing serious complications like intestinal strangulation if surgery is delayed. A multi-national study performed by the National Institute for Health and Care Research proposed that hernia be utilized as a representative 'tracer condition' for the care pathways of other elective general surgery conditions which may also face this complex issue of care timing [5]. As illustrated in this study, a notable delay in care timing was associated with the time before diagnosis [5]. One key factor which influences this period is

when patients decide to engage with formal care, a decision which lies beyond the control of health providers. This study aims to explore how patients approach the navigation and triage of their elective hernia condition within the Kenyan surgical care system.

## Methods

This report presents the qualitative portion of a more extensive multiple-methods study that explored the patient experience accessing surgical care for hernia in Kenya through an interpretivist lens (Fig 1). We applied the Standards for Reporting Qualitative Research (SRQR) checklist to account for all aspects of qualitative research [6] (See S1 Table).

### Study setting

This study occurred at Moi Teaching and Referral Hospital (MTRH), a public tertiary referral hospital in Kenya that serves an estimated patient population of 25 million, with a catchment area covering western Kenya, eastern Uganda, and southern South Sudan [7]. This study was approved by the Moi Teaching and Referral Hospital/Moi University Institutional Research Ethics Committee (MTRH/MU IREC) (IREC/574/2023, Approval 0004517) and the Indiana University School of Medicine Human Subjects & Institutional Review Board (IRB Protocol No. 19278). This study was verbally introduced and written consent was obtained from all patients prior to interview.

### Researcher reflexivity

Members of the study team who directly engaged with patients included a female, American, English-speaking surgery resident physician with a medical degree (HL), a female, Kenyan research coordinator with a nursing degree (SN), and a male, Kenyan undergraduate medical student (JT). HL had over two years of experience working in clinical surgery research at the study site and completed formal qualitative training prior to this study. SN had 10 years of experience working as a research coordinator at the study site, is aware of local contextual characteristics and culture, and acted as an interviewer in numerous qualitative projects at the same study site prior to the initiation of this study. JT spent nine years undergoing medical training at the study site, is fluent in Kiswahili, is aware of local contextual characteristics and culture, and was trained by the study team in qualitative coding methods. JT also attended numerous interviews with SN as a part of his training.

The remainder of the team, composed of expert surgeon scientists and senior qualitative researchers, provided methodological and disease-specific expertise relevant to the study. The principal investigators (ISS, JHS) each had a many-year history of working at the local study site and were able to provide contextual details related to surgical care

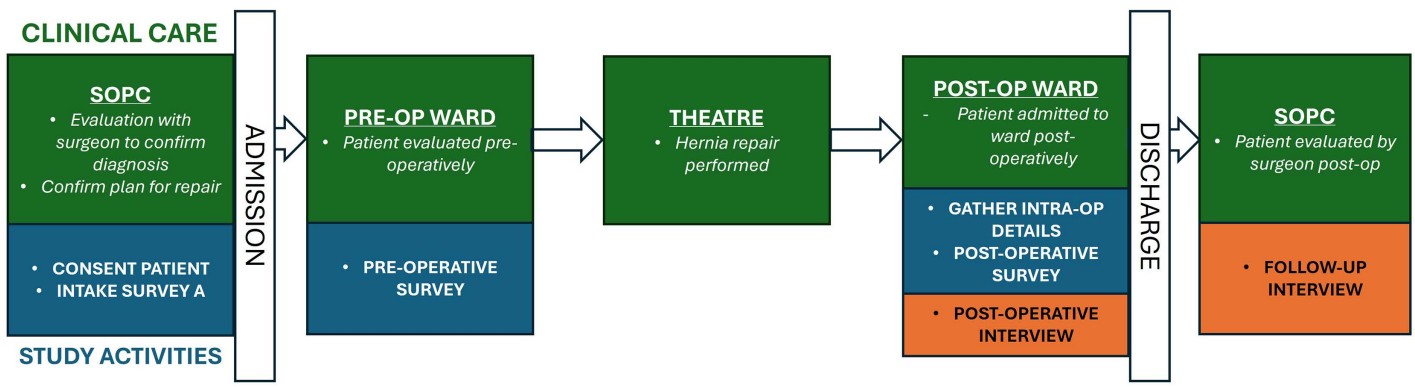

**Fig 1. Procedural diagram of clinical care involved in study activities.**

processes and healthcare administration. TA is a current Kenyan surgical registrar at MTRH who also provided expertise in local provider workflows and patient care processes. These surgeons (TA, ISS, JHS) regularly participated in the care of hernia patients and regularly collaborated with hospital staff who were also involved in this care. One surgeon scientist (JB) is an American hernia specialist who provided the important insight into international standards regarding hernia care. The senior qualitative researchers (SM, RM) had extensive experience in qualitative methodologies, and one (CNN) has significant experience engaging with mixed-methods research at our study site. All members of the study team assisted in reviewing data analysis and manuscript drafts.

## Participant recruitment

We utilized convenience sampling to enroll all adult patients over 18 years of age who presented in-person to the MTRH Surgery Out-Patient Clinic (SOPC) with a diagnosis of hernia. Each diagnosis was confirmed through physical exam by a member of the surgical clinical team prior to enrollment. All patient interactions took place between November 2023 and March 2024.

SN is fluent in Kiswahili and performed the majority of patient interviews to allow patients the flexibility of speaking in English or Kiswahili based on their preference. The interviews took place in a quiet office within the study site which was agreed upon by both interviewer and participant. Family members were allowed to accompany patients during the interview per patient preference. SN and HL were both engaged in guiding patients of through the study background and purpose, taking care to distinguish the purposes of study participation from those of their clinical care and emphasizing that withdrawal from the study at any time would have no implications related to their care. The study team utilized teach-back techniques with patients prior to consent to ensure adequate understanding. Written consent, whether using signatures or thumbprinting in patients with limited literacy, was obtained from all enrolled patients prior to interview. Patients were offered refreshments during interviews but received no other financial incentives.

## Interview guide development

Our interview guide was modeled after the Three-Delay Framework, which describes the three key delays that occur in obstetric care: delays in initiating care seeking, reaching a health facility, and receiving adequate care [8] (see S2 File). This framework provides valuable consideration of not only existing infrastructure, but also patient agency within this infrastructure which is critical when considering care seeking activities for an elective hernia condition. We mirrored this framework to define the three key phases of surgical care seeking for hernia and further expanded specific areas like post-operative recovery to gain an understanding of the overall patient experience with care (Fig 2). To account for the complex referral patterns which patients experienced after initiation of care seeking, we defined the 2nd phase as a patient's first engagement with an external care provider until reaching the center of definitive treatment of the hernia – in this case, MTRH.

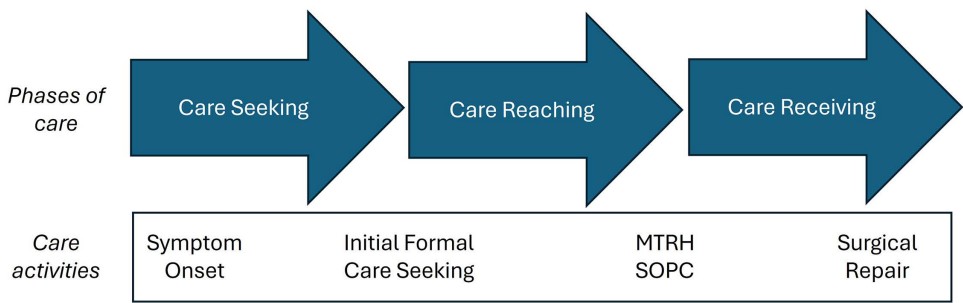

**Fig 2. Care activities within three key phases of surgical care seeking.**

The interview guide was reviewed and translated from English to Kiswahili by professional translators and was piloted with a cohort of eligible patients to ensure clarity and comprehensibility. All interview data, including the pilot interviews, were included in final analysis. Interviews were recorded, transcribed verbatim, translated from Kiswahili to English as needed by an external professional translator service with no direct involvement in the study, and reviewed by SN to ensure accuracy.

## Interview processes

SN completed two semi-structured, in-person patient interviews with patients following hernia repair and following the first post-operative clinic visit (Fig 1). During each interview, the team also guided patients in creating a visual representation of their care journeys which we later refined into "journey maps" to help capture patients' perspectives of their care journeys [9]. Due to limited contact opportunities, interview scripts were not returned for member checking by patients. However, the collaborative process in generating "journey maps" with patients during their interviews allowed patients to provide real-time feedback regarding study team comprehension of their journeys.

We performed audio recordings of all interviews using a recorder device. When patients could not attend both interviews, SN combined interview guides into one extended interview session. Patients who were unable to attend interviews in person were contacted via phone call. Following each interview, SN and HL debriefed to summarize findings, ensure clarity and relevance of interview guide questions, and to assess progress to thematic saturation. Field notes were written during the interview with additional observations added following interview debrief as appropriate.

Of note, a national healthcare worker strike occurred during the data collection period between March to May 2024, delaying patient enrollment due to clinic closures. Following the strike, SN re-engaged previously enrolled patients. Four patients sought surgical repair from private hospitals during the strike, but each remained willing to provide interviews about their care seeking journey.

## Analysis

All audio interviews were de-identified prior to translation and verbatim transcription for analysis. A trained qualitative research transcriber contracted through the AMPATH Qualitative Research Core and was not a part of the study team performed translation and transcription of scripts. The coding team (HL, JT) coded and analyzed all interviews in English using Dedoose (Version 9.0.107), a computer-assisted qualitative data analysis software [10]. The coding team (HL, JT) utilized Braun and Clarke's six-step model of thematic analysis with a contextualist approach [11,12]. Initial deductive codes were derived from the Three Delays Framework, followed by open coding methods through line-by-line coding to generate additional inductive codes [11]. We used direct negotiated agreement between two independent coders to refine all inductively generated codes, resolve disagreements and identify connections to existing deductive codes. Patient "journey maps" were utilized during analysis to provide situational context.

An initial codebook was generated from the first ten interviews which primarily reflected recurring concepts within the data addressing barriers/facilitators to care, patient experience in care and clinical and socioeconomic risk factors. The initial codebook was organized according to the Three Delays Framework phases of care. Each code was assigned a definition of its meaning to provide consistency in future applications. The coding team iteratively reviewed this codebook to generate the final framework consisting of 24 main codes which were clustered to generate overall themes highlighting patient goals, actions, and experiences with care. The final codebook was applied to remaining interviews and data analysis continued until saturation, defined as the absence of newly emerging codes or themes were identified by either independent coder (S5 File). The team determined reaching saturation following analysis of 29 interviews. Verbatim quotations representing themes and subthemes were identified during the coding process and compiled per team consensus during analysis.

## Results

A total of 38 patients were enrolled: 18 patients completed both post-operative and follow-up interviews, and 13 patients completed one combined interview, leading to a total of 49 interviews. Five patients (13%, 5/38) were lost to follow-up and did not complete an interview. With an average of two interviews completed per participant, this resulted in an average of 40 minutes of total interview time per participant. Total patient demographics can be found in Table 1.

We identified three main emerging themes across the three phases of care seeking, reaching, and receiving:

**(1) Power is transferred as consent or knowledge**: Patients desire power to manage their care. Loss of power to control symptoms initiates the need to transfer power to providers through consent. Providers can re-empower patients through providing correct knowledge regarding their condition.

**(2) Trust is the currency for engagement**: Trust is a limited resource and is required for patient engagement and retention in care. Satisfaction with prior experience or influence from trusted contacts can result in gain or loss of trust.

**(3) Context is the foundation for care**: Context, defined by both internal patient factors and external healthcare environment factors, provides the foundation required to make patient-provider care activities possible.

We will further explore these themes below. Each representative patient quote is followed by details about the speaker's study identifier (ID), sex, age, and hernia type. Major themes with subthemes and additional representative quotes are displayed in S3–S5 Tables.

### Power is transferred as consent or knowledge

Centered in the first phase of care seeking, this theme illustrates how patients express desire power over their own health and frequently sought to maintain this power by attempting management of their symptoms independently prior to formal care seeking. Self-management may include utilizing pain medications or self-reducing the hernia swelling when visible.

*"I was shocked that I was feeling pain and when I looked at it, I had swollen. So, I put pressure on it, and it was back to normal"* (ID15, male, 28 years, inguinal)

**Table 1. Patient demographics.**

|  | Total | Male | Female |
|---|---|---|---|
| No. patients, no (%) | 38 | 21/38 (55) | 17/38 (45) |
| Age (years), mean (SD) | 49.6 (18) | 49.8 (19) | 49.3 (16) |
| Age ≤ 50 years, no (%) | 18/38 (47) | 9/21 (43) | 9/17 (53) |
| Age > 50 years, no (%) | 20/38 (53) | 12/21 (57) | 8/17 (47) |
| Hernia type, no (%) |  |  |  |
| Umbilical | 2/38 (5.3) | 0 | 2/17 (12) |
| Supraumbilical | 15/38 (40) | 5/21 (24) | 10/17 (59) |
| Incisional | 5/38 (13) | 1/21 (5) | 4/17 (24) |
| Inguinal (Right) | 14/38 (37) | 14/21 (67) | 0 |
| Inguinal (Left) | 2/38 (5) | 1/21 (5) | 1/17 (6) |
| Employment status, no (%) |  |  |  |
| Formally employed | 5/38 (13) | 3/21 (14) | 2/17 (12) |
| Informally employed | 16/38 (42) | 9/21 (43) | 7/17 (41) |
| Unemployed | 17/38 (45) | 9/21 (43) | 8/17 (47) |

While patients desired the power to manage their own condition, they frequently addressed only the fluctuating symptoms associated with hernia, such as pain or visible swelling, lacking a deeper understanding of the underlying pathology itself.

*"It used to come and go, back the I would just use some painkillers…I never went to the hospital" (ID17, female, 56 years, epigastric)*

With escalation or acute changes in symptoms, patients feel a loss of power which inspired the willingness to transfer of power to healthcare providers in the form of consent.

*"The reason why I went to the hospital, even though at first time I was afraid to go to the hospital, [was that] it reached to a point that the pain got worse, so that is when I decided" (ID8, male, 31 years, inguinal)*

However, the threshold at which patients become willing to transfer power through consent to care may vary. Some patients were quickly willing to consent to formal care, while others delayed until significant functional limitations ultimately left them with no choice but to consent to care.

*"Before it would come [and] I would feel pain for a while, then the pain is done…then I noticed the swelling. I was not able to bend or sits so that's when I told my brother [and] we came to the doctor" (ID21, male, 18 years, inguinal)*

After receiving definitive care, patients realized their own lack of knowledge regarding their disease. Patients expressed how this knowledge would have been instrumental in prompting expedited consent to care.

*"I didn't know which condition it was. If I had known earlier, I would have gone for the ultrasound much sooner and gotten the results so that I can get help" (ID22, male, 46 years, inguinal)*

**Trust is the currency for engagement**

Centered in the second phase of care reaching, this theme shows how patients described trust as an important component which drove patients towards or away from particular care centers or providers, ultimately impacting how they chose to navigate between the numerous care options available. Patients without prior personal experience receiving formal healthcare services often relied on trusted family or community members for guidance about which facilities to invest trust in.

*"When I found out that I had this disease, some of the family members came and advised me to come here…I was told that the best hospital for surgery is [MTRH]" (ID4, male, 57 years, inguinal)*

For patients with prior experience in receiving formal healthcare services, even for conditions outside of hernia, trust was influenced by previous levels of satisfaction with care with specific centers.

*"The previous admissions in the other hospitals I was not prepared nor after even the surgery, I did not go back for the checkup. This one for the experience here has been wonderful different. Regardless of whatever happens in the future I would still recommend MTRH and I would come back here" (ID12, male, 53 years, inguinal)*

Patients expressed the highest satisfaction with providers who appear invested in their well-being. At times, this was even favored over providers who appeared knowledgeable but uncaring.

*"There was a time I went to the hospital, and they said there was nothing… then I thought they were not much concerned… [that is] when I came here [and] I started treatment here." (ID17, female, 56 years, epigastric)*

Once patients engaged with a trusted provider, they often noted a sense of relief with being able to transfer the responsibility for their care to a perceived expert who had their best interests in mind, even if patients themselves still lacked complete knowledge regarding their care.

*"[The surgeons] told me that it won't be painful while they're operating on me. All I had to do was relax and let them do what they studied for and been doing for a long time and make sure that I'm in a safe state. I gave them their time to do their work" (ID28, male, 26 years, inguinal)*

However, inadequate trust in a provider or health center can lead to patients disengaging with care and seeking care elsewhere.

*"At [regional clinic], they tested me and [I] paid 10,000 KSH. Even after that there was no change. That when I went to Referral hospital in Eldoret"* (*ID14, female, 70 years, epigastric*)

## Context is the foundation for care

Centered in the final phase of care receiving, this theme showed how once patients recognized the need to seek care and had sufficient funds of trust to engage with care, they were then required apply the factors of the existing care context around them. Patients prioritize their health and are willing to sacrifice both time and money to reach ideal care. However, the context of patients' internal capabilities and their external local healthcare environment can limit their ability to get the care they desire.

*"I told the doctor that there was something disturbing me, so he asked me to come in and show him. In there I took off my trousers and showed him. He told me that he was sorry, and the condition was called hernia. He told me the only place to go to was Referral. I asked him to help me since he was a doctor, I couldn't go far while he was nearer. He told me he can't" (ID5, male, 78 years, inguinal*)

Patients described the characteristics of ideal care as being close in distance, affordable, high-quality, and efficient.

*"[At referral] everything is there for you, there are machines, there are specialists, it is not a place to guess"* (*ID23, male, 42 years, epigastric)*

Patients will prioritize their own health and are willing to sacrifice time and money in an effort to reach this ideal care.

*"I had a few challenges [with] money and transport…I had bought some iron sheets to build a house, but I had to sell them since health is more important"* (*ID18, male, 30 years, inguinal*)

However, factors internal to a patient, including financial or material resources, may limit in their ability to reach this ideal care.

*"Yes, they asked me visit District Hospital, but I didn't have the money to travel. So, I resorted to getting medicine from a dispensary because it was closer from home"* (*ID16, female, 55 years, epigastric*)

Many patients expressed beliefs in more 'ideal' care being present in private care facilities, sometimes founded in personal experiences with external limitations in local public hospitals, whose inadequate material or human resources also prevent patients from attaining ideal care.

*"[Other hospitals] don't own any machinery that can be used for treatment"* (*ID9, male, 63 years, inguinal*)

Yet many patients facing internal financial limitations expressed the need to rely on public health facilities due to the prohibitive costs of private facilities.

*"I was asked to book for clinic at the public hospital in Kitale, but I opted for a private hospital, of which I was requested to pay for 40,000 Kenyan shillings despite having NHIF. This prompted me to look for another option" (ID5, male, 78 years, inguinal)*

### Integrated conceptual framework

Throughout the patient's care-seeking journey, we note the key themes of power transfer during care, trust in providers fueling engagement, and the need for adequate contextual factors, both internal and external to a patient, to facilitate care. While these three themes were centered in the three phases of care seeking, reaching and receiving, considering these themes more broadly show how these elements do not function in isolation and often influence the others. In this framework, we depict the relationships between these factors as a machine which aims to generate the appropriate care which patients need, in this case, for management of their hernia condition (Fig 3).

**Flow of power:** *Transfer between patients and providers.* Within the machine itself, we depict patients and providers as individual cogs. These two elements interact and affect each other's movements through intermediate engaging forces representing consent and knowledge. To generate care, a patient must engage with and transfer power to the provider via consent. While this transfer may be viewed as a loss of power to the patient, it should instead stimulate a cyclical flow of power whereby the provider cog re-empowers the patient through the transfer of accurate clinical knowledge. This accurate knowledge may then strengthen patient's ability to pursue appropriate care. However, inaccurate knowledge would not encourage this strengthening effect and may result in delayed patient consent to care. Of note, the consent element is critical to the engagement of patients with providers. The threshold for willingness to consent may vary between patients and external influence from trusted companions may assist in facilitating a patient's willingness

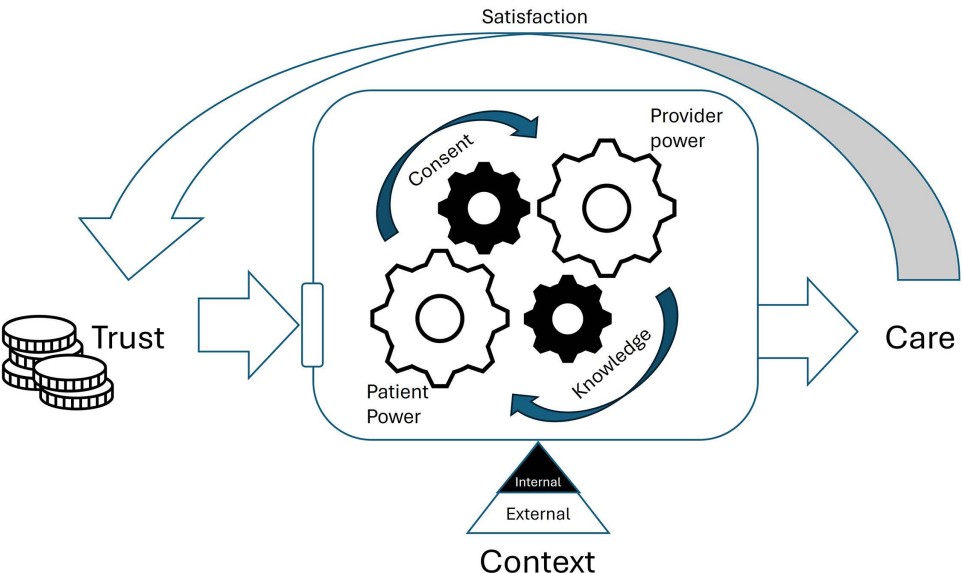

**Fig 3. Conceptual framework integrating three main emerging themes.**

to consent as well. Without consent, the forward movement to generate care is impossible, regardless of the extent of knowledge which providers may provide to patients.

**Funds of trust:** *A limited patient supply.* We depict trust as a resource of funds spent to initiate the engagement of patients and a specific provider or care center, further stimulating the flows of power and generation of care. These funds are limited and can be gained or lost based on a patient's personal satisfaction with care or on the influence of trusted contacts. Satisfaction with care may also have a feedback effect, generating either virtuous promoting effects or vicious barrier effects, on future trust in care engagements. With adequate funds of trust, the machine which brings together patients and providers can run, generating the patient-provider interactions needed for care. With absent or inadequate trust, patients may disengage with the provider and depart to invest trust in another source.

**Context as the foundation:** *Contributing internal and external factors.* Lastly, we depict the overall "machine" balanced on the foundational context of both internal patient factors and external environmental limitations. Internal patient factors may include personal factors which impede patients' ability to reach ideal care, including financial or material resources. Thus, public hospitals like MTRH remain an important, affordable option for quality care for financially vulnerable patients. External environmental limitations, including a lack of accessible hospitals, adequate equipment or human resources, can also limit the care which is accessible by patients. Surgical care is arguably more affected by external environmental limitations due to the need for a required minimum level of resources to make surgery possible – namely, surgical theatres, supplies, and multi-disciplinary peri-operative staff. Both internal and external contextual inadequacies are often impossible for patients to address immediately or may even be entirely outside of patients' control. However, the lack of foundational context may cause the collapse of all patient and provider interactions.

## Discussion

Our study places many familiar factors required for provision of surgical care across the stages of care seeking, reaching and receiving into a framework which is arguably unique in its cyclical nature and its flexibility for use across patient-provider interactions at varying care levels.

The diversity of Kenya's six-level care system is well reflected in the variety of providers which patients referenced within our interviews, ranging from local chemists or pharmacies, to national referral centers [13,14]. Each of these care levels is staffed by providers with differing backgrounds, training, and access to resources, all of which impact the effectiveness of care this provider can give. Given that surgical capabilities in Kenya emerge only at Level 4 sub-county systems and above, a patient's ability to access the appropriate level of care for hernia depends on referral following a correct diagnosis of this surgical condition, often by lower-level local centers. Thus, many patients will engage with complex care-seeking decisions before finally arriving at a center with the capacity to treat their hernia. However, key factors of patient power and patient-provider trust influences how both patients and providers interact with the existing care context.

As patients initiate their care journeys, they frequently have limited insight into larger systemic challenges which providers and administrators must manage. Inadequate personnel on a busy surgical service may be experienced as inattentive providers, and broken imaging equipment may be viewed as a low-quality hospital. These instances of 'poor quality' care lead to decreased funds of trust in that care site or provider which lead to patients utilizing their power or agency to depart and seek that ever elusive 'ideal' care elsewhere. Morris et al. noted compliance with referrals for maternal and neonatal complications in Somalia also depended on trust in medical authority, which may be earned through previous personal experiences or experiences of trusted peers [15].

Trust also becomes an important contributing factor to "path dependence" in patient care journeys, whereby the influence of a current situation can "importantly and persistently shape the situation in the future" [16]. As shown in our interviews, a patient's positive or negative experiences during care seeking at one hospital can affect their perceptions of that care center for all future health needs – a belief which can not only shape that individual's care seeking decisions for their own care, but also that of others for whom the patient is considered a trusted reference. Gakunga et al. reports

how patient-provider communication in management of early-stage breast cancer played a major role in patient satisfaction and willingness to consent to care [17]. Similar patterns are also found in high-resourced settings like the US, where historically underserved Black patient populations reported lower levels of trust and communication compared to White patients [18,19]. This barrier to care resulted in significantly higher time to surgery despite being in a high-resource setting [18,19]. These studies emphasize the influence of 'power' and 'trust' on how patients interact with existing care infrastructure regardless of resource level. However, in the Kenyan setting, where fewer barriers exist to limit patient power or agency in directing self-referral activities, how patients seek care based on their values may have particularly important implications for the balance, or lack thereof, of patient flows and demand across different care centers.

From the provider lens, this degree of patient agency may be experienced as disorganized patient self-referrals which generate imbalances in supply and demand for already stressed surgical care systems [20]. Albutt et al. describes the challenges which providers face, including limited materials, equipment, personnel or capacity, and Pittalis et al. notes the consequences of absent lower-level surgical centers which then drive patients to seek care at busy high-level referral centers [20,21]. The persistent lack of demand for care at lower-level centers may then lead to continued challenges with maintaining adequate funding, staffing or material resources.

Our patients defined 'ideal' care as close in distance, affordable, high-quality, and efficient. However, all interviewed patients ultimately received care at a national referral hospital following an often winding journey through the six-tier care system in Kenya, interacting with providers of all backgrounds who helped to guide their progress. At each branchpoint, this framework incorporating the funds of trust, the transfer of power, the relevant context and the satisfaction in care outcomes can be applied. The alignment of each of these factors can lead to generation of care – in this case, repair of a simple hernia – while breakdown of the 'machine' led to delays and disengagement with care. Thus, each interaction with a provider, whether at a local pharmacy or a referral hospital, is an opportunity to not only shape a patient's care seeking behavior for their hernia, but arguably for future surgical needs as well. Identifying not only inefficiencies in the 'machine', but also at which step of the care cascade these breakdowns occur, become important when considering the needs of elective hernia care.

Elective surgeries like hernia repairs allow time for consideration of numerous care centers or providers before patients decide to transfer power and consent to care, compared to emergent surgeries which remove this patient agency. Prin et al. notes the importance of comparing rates of elective and emergent surgeries as an indicator for surgical access and capacity, reflecting not only the structural capacity to provide adequate surgical care, but also patients' ability to access this available care [22]. There are well-known contextual challenges to surgical care-seeking and provision which have been documented in literature. Considering individual context, a survey study done by Bijlmakers et al, in Malawi noted that despite the absence of direct costs for care services in public hospitals, patients continued to struggle with substantial non-medical direct or indirect costs associated with accessing care, including travel costs or lost income, particularly in lower income groups [23]. In the setting of such limited financial resources, the importance of an efficient process to reach appropriate care gains an even higher importance. Therefore, rather than attempting to expand the number or scope of care systems, thus adding additional complexity, strengthening the patient-provider trust relationship in the existing external care systems may help to shape how patients choose to utilize their power to engage with the care available in their context, thus maximizing the care which is accessible and acceptable for patients.

## Limitations

In our study, all patients were enrolled through the SOPC clinic at a single national referral hospital. Therefore, we recognize the sampling bias associated with only enrolling patients who successfully reached our institution for care. We believe, however, that the complexity of patient journey maps had a protective effect against these biases by illustrating the successes and challenges which patients experienced at different levels of care. From this, we were able to draw general patient desires towards care, standards which could equally be applied to our institution as well.

We also interviewed patients following their successful hernia repair at our site which can introduce recall bias into their perception of both the care they received at our center as well as the care they may have failed to receive at other centers. However, given that we were seeking to capture overall patient care-seeking journeys and explore the factors which influence patient care decision making, we felt that the biased perceptions patients may carry could illustrate the key factors which were most vivid to patients in their care journey, thus further highlighting the significance of the forces named in our model.

## Conclusion & next steps

Our framework describes the role of power, trust and context on patient care seeking behavior and illustrates how these factors interact and influence each other. As described in systems science, "the behavior of a complex system as a whole cannot be easily determined from consideration of its parts individually" [16]. We hope to expand on our current work by applying this framework to patient experiences at different level care centers to expand representation of experiences. We also hope to consider implementation science frameworks such as the COM-B model, which describes the capability, opportunity, and motivation which contribute to behavior change, to help explore how to positive influence patient care seeking behaviors [24]. Acknowledging patient agency, influenced by power and trust, within existing surgical infrastructure, or context, may positively influence the success of the implementation of any larger systemic efforts or policies to improve access to elective surgical care.

## Supporting information

**S1 Table. Standards for Reporting Qualitative Research (SRQR).**
(DOCX)

**S2 File. Semi-structured interview guide.**
(DOCX)

**S3 Table. Table 1. Patients desire power over care: Theme definition, subthemes, and representative quotes.**
(DOCX)

**S4 Table. Table 2. Trust is the currency for engagement: Theme definition, subthemes, and representative quotes.**
(DOCX)

**S5 Table. Table 3. Context of healthcare environment remains foundation for accessing care: Theme definition, subthemes, and representative quotes.**
(DOCX)

**S4 File. PLOS Inclusivity in global research questionnaire.**
(DOCX)

**S5 File. Codebook utilized in qualitative analysis.**
(DOCX)

## Acknowledgments

The authors thank the patients who participated in our study, as well as all staff and surgeons at Moi Teaching and Referral Hospital (MTRH), Moi University School of Medicine (MUSM), and the Academic Model Providing Access to Healthcare (AMPATH) who supported the study.

## Author contributions

**Conceptualization:** Helen W. Li, Teddy Ashibende Aurah, Jeffrey A Blatnik, JoAnna Hunter-Squires, Ivan Seno Saruni.

**Data curation:** Helen W. Li, Charles Nathan Nessle, Ivan Seno Saruni.

**Formal analysis:** Helen W. Li, Charles Nathan Nessle, Ivan Seno Saruni.

**Funding acquisition:** Helen W. Li, Jeffrey A Blatnik, JoAnna Hunter-Squires, Ivan Seno Saruni.

**Investigation:** Helen W. Li, Jesse Kiprono Too, Sarah Nyanchama Nyariki, Teddy Ashibende Aurah, JoAnna Hunter-Squires, Ivan Seno Saruni.

**Methodology:** Helen W. Li, Sarah Nyanchama Nyariki, Charles Nathan Nessle, Sara Malone, Rachel Matsumoto, JoAnna Hunter-Squires, Ivan Seno Saruni.

**Project administration:** Helen W. Li, Jesse Kiprono Too, Sarah Nyanchama Nyariki, JoAnna Hunter-Squires.

**Resources:** Helen W. Li.

**Software:** Helen W. Li.

**Supervision:** Helen W. Li, Jeffrey A Blatnik, JoAnna Hunter-Squires, Ivan Seno Saruni.

**Validation:** Helen W. Li, Jesse Kiprono Too, Sarah Nyanchama Nyariki, Sara Malone, Rachel Matsumoto, Teddy Ashibende Aurah.

**Visualization:** Helen W. Li, Jesse Kiprono Too, Sarah Nyanchama Nyariki, Rachel Matsumoto, JoAnna Hunter-Squires, Ivan Seno Saruni.

**Writing – original draft:** Helen W. Li.

**Writing – review & editing:** Helen W. Li, Jesse Kiprono Too, Sarah Nyanchama Nyariki, Charles Nathan Nessle, Sara Malone, Rachel Matsumoto, Teddy Ashibende Aurah, Jeffrey A Blatnik, JoAnna Hunter-Squires, Ivan Seno Saruni.

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
