## [Decision Letter · Decision Letter 0]

11 Jun 2025

Dear Dr. Li,

Thank you for submitting your manuscript to PLOS ONE. After careful consideration, we feel that it has merit but does not fully meet PLOS ONE’s publication criteria as it currently stands. Therefore, we invite you to submit a revised version of the manuscript that addresses the points raised during the review process.

We look forward to receiving your revised manuscript.

Kind regards,

Lovenish Bains, MS, FNB, FACS, FRCS (Glas), FICS, FIAGES

Academic Editor

PLOS ONE

Journal Requirements:

3. Thank you for stating the following in the Acknowledgments Section of your manuscript: [Research reported in this publication was supported by the Fogarty International Center of the National Institutes of Health under grant #D43TW009345 awarded to the Northern Pacific Global Health Fellows Program. The content is solely the responsibility of the authors and does not necessarily represent the official views of the National Institutes of Health. The authors thank the patients who participated in our study, as well as all staff and surgeons at Moi Teaching and Referral Hospital (MTRH), Moi University School of Medicine (MUSM), and the Academic Model Providing Access to Healthcare (AMPATH) who supported the study.]

Please remove any funding-related text from the manuscript and let us know how you would like to update your Funding Statement. Currently, your Funding Statement reads as follows: “The authors received no specific funding for this work.”

4. In the online submission form, you indicated that [De-identified textual data can be made available upon reasonable request to the corresponding author.].

Reviewers' comments:

Reviewer's Responses to Questions

**Comments to the Author**

1. Is the manuscript technically sound, and do the data support the conclusions?

Reviewer #1: Yes

Reviewer #2: Yes

Reviewer #3: Yes

Reviewer #4: Partly

Reviewer #5: Yes

2. Has the statistical analysis been performed appropriately and rigorously?

Reviewer #1: N/A

Reviewer #2: N/A

Reviewer #3: N/A

Reviewer #4: I Don't Know

Reviewer #5: N/A

3. Have the authors made all data underlying the findings in their manuscript fully available?

Reviewer #1: No

Reviewer #2: Yes

Reviewer #3: Yes

Reviewer #4: Yes

Reviewer #5: Yes

4. Is the manuscript presented in an intelligible fashion and written in standard English?

Reviewer #1: Yes

Reviewer #2: Yes

Reviewer #3: Yes

Reviewer #4: Yes

Reviewer #5: Yes

Reviewer #1: This qualitative study explores how Kenyan patients diagnosed with elective hernias navigate surgical care based on interviewing 38 patients.

The three main themes presented include: 1.power transfer between patients and providers through consent and knowledge, 2. trust as a limited currency that governs engagement, 3. contextual internal/external factors that shape the care pathway.

This study highlights patient-centred barriers to surgical care in low resource setting and demonstrates depth and authenticity employing qualitative data from interviews. Having Kenyan collaborators improves contextual sensitivity and ethical conduct.

However, selection and recall bias may affect findings as all the patient were interviewed after accessing and receiving tertiary care, overlooking those who never reached care. This creates a skewed perspective. The "machine" metaphor and cyclical model may overinterpret what are fairly intuitive findings and adds unnecessary complexity without improving clarity or utility.

Reviewer #2: Thank you for the opportunity to review this thoughtful work. The authors describe their semiquantitative work in a survey of patients undergoing hernia surgery in a referral hospital in Kenya. The writing is intelligible, and experimental design is appropriate. As someone less versed in the background of this work, I sometimes struggled to understand pieces of the discussion.

I appreciate the reference to the three delays in OB care. I think additional discussion of the difference in urgent OB care and elective hernia care could be expanded upon. Additionally, my humble experience in this setting is that cost is the major driver of a decision to recieve hernia surgery. This is only indirectly addressed. It might be worthwhile expanding on this as well.

Figures were appropriate and clear

Overall, the manuscript could be shortened by 10-15% without affecting the clarity of the presentation.

Reviewer #3: The authors present an interesting article looking at the factors that contribute to hernia care in Kenyan patients. The authors found that the flow of power and trust interactions within in infrastructure affected by external and internal context to affect the care and satisfaction of patients following hernia surgery in Kenya.

These concepts are well-understood throughout the medical community when it comes to informed consent within the USA. How do these concepts differ in a resource lacking country such as Kenya? Does this apply broadly to any resource stricken country?

Context is obviously very important. Does the context of the majority of authors/researchers being from the USA affect how the Kenya patients react to such questions regarding their care vs. researchers from Kenya?

What was the average education level of the Kenyan patients? This would be important to know to better understand how the trust/power dynamics differ in a country with such different educational standards than the USA.

The main outcomes of these flows of power should lead to improved outcomes/satisfaction, but the authors do not include any data regarding outcomes/satisfaction in this study. Is this data available and able to be included? This would greatly improve the impact of this study.

Reviewer #4: I would like to recuse myself from this, due to insufficient background in public health. The study is based on an interview process and quotes utility of a model I am not familiar with. The study has sociological implications and requirements which I would not be an authority on to review.

Reviewer #5: I would like to commend the authors on a well-written and insightful paper. Research on access to care often focuses on health systems or provider perspectives, so it is refreshing and commendable that this study centers the experiences of patients themselves.

Specific Comments:

• SRQR Checklist: I appreciate the team’s adherence to the SRQR checklist and commend the thoroughness of the reporting.

• Informed Consent (Lines 91–92; 127–128): The manuscript notes that patients were verbally introduced to the study and provided written consent. Were all participants literate, and did they fully understand the implications of signing a consent form? In contexts where literacy may be limited, it is common to use thumbprints as an alternative. It would be helpful to include a brief note on any accommodations made for participants who were not literate.

• Voluntariness of Participation (Lines 125–126): In hierarchical healthcare settings, distinguishing between voluntary research participation and routine care can be challenging. What measures were taken to ensure that patients did not feel coerced into participating, particularly out of concern that refusal might affect their care? Additionally, did any patients decline to participate? If so, were reasons for refusal documented?

• Journey Maps: The use of collaboratively created “journey maps” is a compelling methodological choice. It would be valuable to elaborate on the types of insights these maps revealed. For instance, did they depict geographical journeys (e.g., from rural villages to urban referral hospitals) or focus more on the care pathways and challenges within Kenya’s six-tier health system? Were there patterns in how patients navigated levels of care—for example, did those accessing private care bypass certain levels?

• Private Sector Insights (Lines 159–161): For patients who sought care in private hospitals, what specific insights emerged? Did their experiences differ significantly from those in the public system?

• Typographical Note (Lines 154–155): There appears to be a missing word in the sentence: “Field notes were written during the interview.”

• Demographic Data (Lines 190–191, Table 1): The demographic distribution by age and gender is well-balanced. However, was any additional socioeconomic or educational data collected? This information could enrich the analysis, particularly in relation to the “three delays” framework. The first delay—deciding to seek care—is often influenced by health literacy and socioeconomic status. While the interview guide (S2_file, Question 7) asks patients to describe a hernia in their own words, it would be insightful to explore how understanding varied by educational background. This is especially relevant given the subtheme “Patients regret lacking adequate knowledge” (S3 Table 1).

• Conceptual Framework (Lines 306–350): The conceptual framework is a strong contribution. I particularly appreciated the depiction of power as a form of “currency” exchanged between patients and providers. The framework’s recognition of both internal and external contextual factors adds depth and nuance to the analysis.

• Typographical Note (Lines 392–393): There is a typo in the phrase “Elective to emergency surgeries.”

This is a well-executed and thoughtful paper that offers valuable insights into how patients navigate healthcare systems in low- and middle-income countries.

**Do you want your identity to be public for this peer review?** For information about this choice, including consent withdrawal, please see our Privacy Policy

Reviewer #1: No

Reviewer #2: No

Reviewer #3: No

Reviewer #4: No

Reviewer #5: **Yes: ** Michael Bahrami-Hessari

---

## [Author Response · Author response to Decision Letter 1]

12 Jul 2025

Comments from Peer Reviewer #1:

This qualitative study explores how Kenyan patients diagnosed with elective hernias navigate surgical care based on interviewing 38 patients. The three main themes presented include: 1. power transfer between patients and providers through consent and knowledge, 2. trust as a limited currency that governs engagement, 3. contextual internal/external factors that shape the care pathway.

This study highlights patient-centered barriers to surgical care in low resource setting and demonstrates depth and authenticity employing qualitative data from interviews. Having Kenyan collaborators improves contextual sensitivity and ethical conduct.

However, selection and recall bias may affect findings as all the patients were interviewed after accessing and receiving tertiary care, overlooking those who never reached care. This creates a skewed perspective. The "machine" metaphor and cyclical model may overinterpret what are fairly intuitive findings and adds unnecessary complexity without improving clarity or utility.

Response: Thank you for this thoughtful feedback. We agree about the risk for selection and recall bias, as noted in our limitations section. We attempted to mitigate the selection bias of enrolling patients who arrived to our referral center by considering the entirety of the patient journey through comprehensive journey maps which patients generated. In these maps, we were able to glean the mixed positive and negative experiences of patients in regional hospitals and referral hospitals alike. While it is true that patients would predominantly report positive experiences with MTRH given they were able to successfully reach care here, they also noted specific aspects of desired care which was not met at regional centers (ex: adequate attention, available resources). These general expectations may still be applied to MTRH as well. Additionally, failure to receive care at other centers offers the perspective behind patients who ‘never reached care’ relative to that specific center. We believe the factors influencing care seeking as illustrated in this framework were drawn from adequately complex care journeys which were still able to capture general patient expectations and desires for care in addition to the consequences of whether those desires were able to be met. We do hope to expand to include patients from a greater variety of centers, particularly of local regional centers, in future works to confirm our findings in this study. Additional text was added to our limitations section and conclusion/next step section relaying these thoughts (Lines 438-444, lines 446-458).

Regarding recall bias, we believe that patient recall bias of their journey may actually emphasize the pieces of the journey which were particularly important to them, offering a unique insight into how they perceive care. We assumed these biases may be diluted across the larger cohort of patient interviews, and we would adequately be able to identify common experiences across all patients. Additional text was added to our limitations section relaying these thoughts (Lines 438-444).

We also appreciate that our framework highlights seemingly intuitive factors which influence care seeking: power, trust and context. However, we hope that it is how our framework places these factors in relation to each other that is more novel. Particularly within the context of a larger systems perspective, it is important in systems science to identify the presence of virtuous or vicious cycles which promote or prevent progress in care seeking, as these can be significant points at which intervention could significantly impact patterns of care. Within these lines, we hope that drawing attention to the cyclical influences which these three main forces have on each other may also highlight areas opportunities for potential interventions on a systems level, as well as potential consequences which changes on one factor may have on another. The cyclical relationship between power and knowledge is described in the ‘flows of power’ section. We added additional descriptions of the cyclical relationship between trust and satisfaction in the ‘funds of trust’ section for clarification (Lines 340-348).

Comments from Peer Reviewer #2:

Thank you for the opportunity to review this thoughtful work. The authors describe their semiquantitative work in a survey of patients undergoing hernia surgery in a referral hospital in Kenya. The writing is intelligible, and experimental design is appropriate. As someone less versed in the background of this work, I sometimes struggled to understand pieces of the discussion.

I appreciate the reference to the three delays in OB care. I think additional discussion of the difference in urgent OB care and elective hernia care could be expanded upon. Additionally, my humble experience in this setting is that cost is the major driver of a decision to receive hernia surgery. This is only indirectly addressed. It might be worthwhile expanding on this as well.

Figures were appropriate and clear. Overall, the manuscript could be shortened by 10-15% without affecting the clarity of the presentation.

Response: Thank you for highlighting the contrast between emergent/urgent OB care and elective hernia care. We believe the elective nature of hernia disease is unique in that it balances with the risk of progression of hernia disease acute bowel incarceration or strangulation which would then transform the situation into an urgent/emergent case as well. However, hernias are also unique in that the only cure involves surgical repair, so patients must actively choose to seek surgical care, or else they will remain with the hernia forever. We added text in the ‘data collection’ section (Lines 137-142) to specify how the Three Delays Framework uniquely considers not only the existing care infrastructure, but also the patient agency in navigating this infrastructure. This is particularly important given our particular interest in patient agency in navigating care for an elective hernia, compared to the diminished patient agency present in urgent/emergent situations as we posit in our discussion.

We also appreciate the insightful comment regarding the impact of healthcare costs on the ability to receive care. We would consider this to be a part of the ‘external context’ of the framework. We particularly noted the importance of cost related to how patients rely on public hospitals as an affordable option for quality care in the ‘context as a foundation’ section. We also added additional text and citation in our discussion section highlighting the importance of developing an efficient process to reach care given the high rates of non-medical direct and indirect costs associated with seeking surgical care as shown in other studies (Lines 369-375).

Thank you for your feedback on the length of this manuscript. We have again reviewed the document to attempt to appropriately refine the length.

Comments from Peer Reviewer #3:

The authors present an interesting article looking at the factors that contribute to hernia care in Kenyan patients. The authors found that the flow of power and trust interactions within in infrastructure affected by external and internal context to affect the care and satisfaction of patients following hernia surgery in Kenya. These concepts are well-understood throughout the medical community when it comes to informed consent within the USA. How do these concepts differ in a resource lacking country such as Kenya? Does this apply broadly to any resource stricken country?

Response: Thank you for this very insightful comment. Having the perspectives of US and Kenyan researchers during analysis of this project exactly highlights how similar concepts were identified from the reports of Kenyan patients as from US patients. By building our framework on the foundation of context, both external, we hope our framework will allow the flexibility to accommodate differently resourced environments. As mentioned at the end of our ‘context as foundation’ section, the absence of an adequate foundation context, a higher risk in lower resourced settings, may result in the collapse of all patient and provider interactions. We added text in our discussion section to highlight this fact again in addition to emphasizing the need to incorporate an understanding of patient agency in any future foundational changes (Lines 385-387, 423-429).

Comments from Peer Reviewer #3:

Context is obviously very important. Does the context of the majority of authors/researchers being from the USA affect how the Kenya patients react to such questions regarding their care vs. researchers from Kenya?

Response: Thank you for this important point. As mentioned in our reflexivity statement, we are proud to note that our team who directly engaged with patients was well balanced, with one US author (HL) working closely with a Kenyan research coordinator (SN) who performed all interviews in Kiswahili/English based on patient preferences, and with another Kenyan author (JT) on all coding activities. Our two Principal Investigators also represented distinct US and Kenyan perspectives, both with significant on-the-ground experience at the study setting. The remaining US authors provided methodological and disease-specific expertise throughout the project. We clarified this point in our reflexivity statement further (Lines 108-119).

As mentioned in our methodology, we piloted the interview guide with local patients to ensure clarity and acceptability of questions. Patients also responded well to using the journey maps as a visual representation of their reports which allowed for real time confirmation of the accuracy of the team’s documentation of patient experiences.

Comments from Peer Reviewer #3:

What was the average education level of the Kenyan patients? This would be important to know to better understand how the trust/power dynamics differ in a country with such different educational standards than the USA.

Response: Thank you for this insightful comment. While we unfortunately did not collect data regarding education level for patients, we did collect information involving employment type may also affect trust/power dynamics. We added data illustrating employment status (formal, informal or unemployed) into Table 1. Through this, we do note that our center serves a vulnerable population with high rates of informal or unemployment. In some ways, this ‘vulnerability’ contrasted with the discussions of patient agency and flows of power emphasize the strong desire amongst all patients, regardless of background, to demand high-quality care.

Additional discussion regarding how education is related to medical literacy, another important factor into trust/power dynamics, can be found below in response to a further reviewer comment: “Demographic Data (Lines 190–191, Table 1)”.

Comments from Peer Reviewer #3:

The main outcomes of these flows of power should lead to improved outcomes/satisfaction, but the authors do not include any data regarding outcomes/satisfaction in this study. Is this data available and able to be included? This would greatly improve the impact of this study.

Response: Thank you for this recommendation. Due to the limitations in word count, we felt it adequate to describe this framework in this manuscript. We did attempt to provide qualitative data on how trust and satisfaction were closely related in the “Trust is a currency for engagement” section (see: satisfaction with prior care increased trust, greater satisfaction with more trusted providers, loss of satisfaction results in loss of trust). This is also illustrated in our framework as the arrow associated with satisfaction in care feeds back to the funds of initial trust. Additional text was added to further clarify this point in our ‘funds of trust’ section (Lines 343-345).

However, we did not include quantitative data measuring patient outcomes or satisfaction as we believed this to exceed the purpose of introducing the framework at this time. We do hope to apply this framework to how these factors can be seen in real patient care flows across our care system in the future, with one anticipated outcome measure including timeliness of patient completion of care. As such, we hope to have more adequate space to better address this important question in a subsequent work.

Comments from Peer Reviewer #4:

I would like to recuse myself from this, due to insufficient background in public health. The study is based on an interview process and quotes utility of a model I am not familiar with. The study has sociological implications and requirements which I would not be an authority on to review.

Response: Thank you for your time and consideration of our work.

Comments from Peer Reviewer #5:

I would like to commend the authors on a well-written and insightful paper. Research on access to care often focuses on health systems or provider perspectives, so it is refreshing and commendable that this study centers the experiences of patients themselves.

Response: Thank you for this kind feedback.

Specific Comments:

1. SRQR Checklist: I appreciate the team’s adherence to the SRQR checklist and commend the thoroughness of the reporting.

Response: Thank you for this kind feedback.

2. Informed Consent (Lines 91–92; 127–128): The manuscript notes that patients were verbally introduced to the study and provided written consent. Were all participants literate, and did they fully understand the implications of signing a consent form? In contexts where literacy may be limited, it is common to use thumbprints as an alternative. It would be helpful to include a brief note on any accommodations made for participants who were not literate.

Response: Thank you for this helpful feedback. SN and HL guided patients through the study process and consent documents prior to consent confirmation. We also asked patients to use the teachback technique to ensure adequate understanding of study processes. The majority of patients were literate and able to sign the document. We did have the alternative for fingerprinting for patients to express consent, but this was very infrequently utilized. We have included these additional nuances in the data collection section for clarification (Lines 132-135).

3. Voluntariness of Participation (Lines 125–126): In hierarchical healthcare settings, distinguishing between voluntary research participation and routine care can be challenging. What measures were taken to ensure that patients did not feel coerced into participating, particularly out of concern that refusal might affect their care? Additionally, did any patients decline to participate? If so, were reasons for refusal documented?

Response: Thank you for this important point. As mentioned in the data collection section, SN and HL took care to distinguish the study participation from clinical care activities when consenting patients. As a part of our ethical approval, we also made sure to emphasize that withdrawal from the study at any time would have no impact on their clinical care. This has been added to the data collection section (Lines 129-132).

Some patients declined to participate due to expressed limitations in time or other pending appointments at the time of enrollment. These patients were not enrolled, and no patient information was tracked or collected to minimize unnecessary risk.

4. Journey Maps: The use of collaboratively created “journey maps” is a compelling methodological choice. It would be valuable to elaborate on the types of insights these maps revealed. For instance, did they depict geographical journeys (e.g., from rural villages to urban referral hospitals) or focus more on the care pathways and challenges within Kenya’s six-tier health system? Were there patterns in how patients navigated levels of care—for example, did those accessing private care bypass certain levels?

Response: Thank you for this encouraging feedback. As mentioned above, due to the limitations in word count, we felt it adequate to describe this framework in this manuscript. Elaborating further on the specific insights of the journey maps exceeded the scope of our current intended purpos

---

## [Decision Letter · Decision Letter 1]

3 Sep 2025

Dear Dr. Li,

Thank you for submitting your manuscript to PLOS ONE. After careful consideration, we feel that it has merit but does not fully meet PLOS ONE’s publication criteria as it currently stands. Therefore, we invite you to submit a revised version of the manuscript that addresses the points raised during the review process.

We look forward to receiving your revised manuscript.

Kind regards,

Lovenish Bains, MS, FNB, FACS, FRCS (Glas), FICS, FIAGES

Academic Editor

PLOS ONE

Journal Requirements:

Reviewers' comments:

Reviewer's Responses to Questions

**Comments to the Author**

Reviewer #1: All comments have been addressed

Reviewer #5: All comments have been addressed

2. Is the manuscript technically sound, and do the data support the conclusions?

Reviewer #1: Partly

Reviewer #5: Yes

3. Has the statistical analysis been performed appropriately and rigorously?

Reviewer #1: No

Reviewer #5: N/A

4. Have the authors made all data underlying the findings in their manuscript fully available?

Reviewer #1: No

Reviewer #5: Yes

5. Is the manuscript presented in an intelligible fashion and written in standard English?

Reviewer #1: No

Reviewer #5: Yes

Reviewer #1: This article reports on factors influencing patient-level decision making leading to elective hernia repair. The authors have clearly made an effort to conduct a qualitative study through interviews, incorporating sociological concepts in their interpretation.

However, the study has notable limitations. The sample size is too small to support conclusions that are representative of the broader Kenyan population. Greater detail in the methodology would improve transparency; for instance, it would be helpful if the interview process and guiding questions were explicitly described, rather than relying primarily on direct quotations. While the authors may be fluent in Kiswahili, it is unclear whether medical terms and nuanced concepts were consistently and accurately conveyed in translation. It is also not stated whether a standardized questionnaire was developed in both English and Kiswahili, and whether this was reviewed by professional translators or interpreters.

The discussion and conclusion could be strengthened by explicitly identifying specific factors influencing decision making at each stage: symptom recognition, care-seeking, and surgical repair. For example, pain, symptoms of obstruction, engagement with primary care, financial or time costs, hospital stay, satisfaction with informed consent, and return to work are all relevant dimensions. Relying on vague terms such as “power of trust” reduces both clarity and credibility.

Overall, the study addresses an important topic, but would benefit from clearer methodology, stronger presentation of findings, and more precise interpretation of patient factors.

Reviewer #5: (No Response)

**Do you want your identity to be public for this peer review?** For information about this choice, including consent withdrawal, please see our Privacy Policy

Reviewer #1: No

Reviewer #5: No

---

## [Author Response · Author response to Decision Letter 2]

6 Oct 2025

Comments from Peer Reviewer #1:

This article reports on factors influencing patient-level decision making leading to elective hernia repair. The authors have clearly made an effort to conduct a qualitative study through interviews, incorporating sociological concepts in their interpretation.

Comment 1: However, the study has notable limitations. The sample size is too small to support conclusions that are representative of the broader Kenyan population.

Response 1: Thank you for this insightful feedback. While we agree that this study cannot make conclusions encompassing the entire Kenyan surgical population, we hope this study offers exploratory insight into care initiation patterns within a cohort of patients seeking care for a foundational general surgery condition of hernia.

Qualitative research methodology posits that thematic saturation may be reached at approximately 9-17 interviews, with an average of around 12 interviews. In our cohort, we have exceeded this number and have additionally attempted to ensure fair representation of sample heterogeneity (ie: male=21 vs female=17; age<50=18 vs age>50=20) as well. From these interviews, we were able to achieve thematic saturation of conclusions, around which we based our results. We added additional details regarding how we identified reaching saturation (Lines 197-202).

From our last round of reviewer responses, we addressed the main risks of sampling bias and recall bias which may affect the applicability of our findings. However, we believe there is significance in the richness of patient experiences which we were able to capture through both patient interviews and journey maps which helps to illustrate a deeper understanding of the patient experience beyond a simpler tool like a survey. We did make an additional clarification that an average of 40 minutes of interview time was completed per patient (Line 210-211).

Comment 2: Greater detail in the methodology would improve transparency; for instance, it would be helpful if the interview process and guiding questions were explicitly described, rather than relying primarily on direct quotations. While the authors may be fluent in Kiswahili, it is unclear whether medical terms and nuanced concepts were consistently and accurately conveyed in translation. It is also not stated whether a standardized questionnaire was developed in both English and Kiswahili, and whether this was reviewed by professional translators or interpreters.

Response 2: We appreciate this important point. We had attached in our submission Supporting File 2 (S2 File) in which we present semi-structured interview guide which was used for all interviews. With regards to the interview guide structure, we have clarified the availability of a Kiswahili version of the guide, produced through a professional translation service, which was used based on patient language preferences (Lines 153-155). However, all interviews were completed by an experienced interview who was fluent in both English and Kiswahili. The interview guide was piloted with patients as well to ensure clarity and comprehensibility while minimizing medical jargon. We also added additional details in how our interviews were also translated from Kiswahili to English during transcription by an external professional translator service not otherwise involved in the study (Lines 153-158).

Overall, our interview guide content and discussions are structured around the Three-Delays Framework, as described in Lines 140-149. However, we aimed to explore a broad understanding of the patient’s condition, including their experience in receiving a hernia diagnosis and whether their understanding changed following this diagnosis. Therefore, our hope was to not only gauge the level of correctness of patient understanding, but also their understanding of the disease/process and how this affected their medical decisions. At times, it was in fact a patient’s lack of understanding which shaped their decision processes which is arguably just as important a finding.

We attempted to ground our findings in our data points of patient interviews, as per the Braun and Clarke method (Lines 185-192). The similar presence of codes across multiple diverse patient reporters as shown in Supplemental File 3, tables 1-3, illustrates the state of thematic saturation. We also described the process by which we organized the codes into a codebook which was then applied for subsequent analysis of interviews, as well as the phase of care associated with each main code within our codebook (Supplemental file 5) which we hope can add to process transparency (Lines 187-202).

Lastly, to improve the organization and clarity of our methods section, we added the additional subcategories of “participant recruitment”, “interview guide development”, and “interview processes”.

Comment 3: The discussion and conclusion could be strengthened by explicitly identifying specific factors influencing decision making at each stage: symptom recognition, care-seeking, and surgical repair. For example, pain, symptoms of obstruction, engagement with primary care, financial or time costs, hospital stay, satisfaction with informed consent, and return to work are all relevant dimensions. Relying on vague terms such as “power of trust” reduces both clarity and credibility.

Response 3: As noted in the methods, the final codebook was refined into a framework of 24 codes which highlighted overall themes involving patient goals, actions and experiences with care. These overall themes are often present across the three stages of care: seeking (symptom recognition), reaching (care navigation) and receiving (surgical repair). The dimensions which the reviewer insightfully lists are all critical to a patient journey and are all aspects explored within patient interviews. However, in our framework we propose that it is not an individual dimension which influences decision making, but rather broader patterns which are seen across all three stages which are linked and feedback onto each other in cyclical fashions.

For example, regarding dimensions like obstructive symptoms during the care seeking phase, we see this initial desire in patients to manage their own health, whether by self-treatment with painkillers or reducing the hernia swelling when visible. However, as they lose the ability to adequately manage their symptoms, patients are forced to look outside themselves and start to engage with care options available to them, which were often numerous. This shifting balance of power from inside to outside the individual is shown as patients consider a transfer of power of their own power to the provider through consent to care.

However, the willingness to transfer power is also dependent on dimensions like trust and satisfaction, most illustrated in the care reaching phase. These feed back onto each other and impacted whether patients remained engaged with care to allow for power transfer.

Lastly, in the care receiving phase, patients were positioned to want to seek care (ready to consent) and willing to engage with providers (adequate funds of trust). However, they may remain limited by the context of existing limitations, including both availability of external care systems capable of providing care or internal patient capacity to seek care. This aspect is seen in dimensions like how patients utilize their limited financial capabilities to pay for care, or how they respond to hospitals which are variably resourced to provide care.

The intertwined nature of these overarching forces extends beyond individual dimensions, which is important to recognize when considering how to best influence patient decisions. We have clarified this intertwined nature in the results (Lines 335-342).

Lastly, we have also added short clarifying descriptions to accompany the sub-headings under the integrated conceptual framework.

Comment 4: Overall, the study addresses an important topic, but would benefit from clearer methodology, stronger presentation of findings, and more precise interpretation of patient factors.

Response 4: Thank you for this overall feedback. We hope that by addressing the points above, we have improved the clarity of our work.

---

## [Editor Report · Decision Letter 2]

9 Nov 2025

Exploring factors contributing to patient decision-making in the care journey to elective hernia care in Kenya

PONE-D-25-20411R2

Dear Dr. Li,

We’re pleased to inform you that your manuscript has been judged scientifically suitable for publication and will be formally accepted for publication once it meets all outstanding technical requirements.

Kind regards,

Lovenish Bains, MS, FNB, FACS, FRCS (Glas), FICS, FIAGES

Academic Editor

PLOS ONE
---

## [Editor Report · Acceptance letter]

PONE-D-25-20411R2

PLOS ONE

Dear Dr. Li,

I'm pleased to inform you that your manuscript has been deemed suitable for publication in PLOS ONE. Congratulations! Your manuscript is now being handed over to our production team.

Kind regards,

on behalf of

Dr. Lovenish Bains

Academic Editor

PLOS ONE